# QPSK MMW Wireless Communication System Based On p-i-n InGaAs Photomixer

**Asemahegn Wudu ***[ID]**, Daniel Rozban and Amir Abramovich**[ID]

Department of Electrical and Electronics Engineering, Ariel University, Ariel 40700, Israel;
rozbandaniel@ariel.ac.il (D.R.); amir007@ariel.ac.il (A.A.)
**\*** Correspondence: asiw@ariel.ac.il; Tel.: +972-3-9066389

**Abstract:** Millimeter-wave (MMW) frequencies (30–300 GHz), located between the microwave and infrared (IR), are promising solutions for the increasing demand of high data rate applications, UHD multimedia, HD gaming, security, surveillance, and the emergence of 5G Internet of Things (IoT). In this article, we experimentally demonstrated MMW wireless communication link using InGaAs p-i-n photomixer and commercially available telecom components at W-band (75–110 GHz). The photomixer was excited by two 1.5 μm lasers via standard telecom fiber optics, to generate frequency difference at W-band. QPSK modulated signal transmitted by the photomixer and received horn antenna integrated MMW mixer and analyzed using a spectrum analyzer and Vector Signal Analyzer (VSA) software.

**Keywords:** coherent detection; millimeter-wave; radio-over-fiber; sub-mm wave; photomixer; photonic THz generation; wireless communication; 5G

## 1. Introduction

In recent years, the study of MMW and THz frequency bands has attracted significant attention due to the fast growth of wireless data communication technology [1]. Within ten years, the required data rate of wireless communication expected to be 100 GB/s [2]. As the demand for high data rate increased due to the emergence of mobile devices such as multifunction mobile phones and tablets, UHD TV, HD graphic video games, and real-time data transmissions, lower frequency bands (below 60 GHz) are extremely limited [3]. Therefore, moving to the MMW and THz frequency bands is essential to exploit the available wide bandwidth. However, the lack of availability of high-frequency components and the high cost and complexity of designing these components makes the construction of MMW wireless communication link very challenging.

In order to close the "THz gap" between the RF and the optical spectrum, new techniques and components combining optical and RF components are required [4]. These techniques include Resonant Tunneling Diodes (RTD), electronic sources (frequency multipliers), and the generation of MMW and THz frequencies using the optical photomixing process [4,5]. Different techniques to realize the optical photomixing have been suggested [6]. Each technique is characterized by the improvement it offers regarding bandwidth, maximum output power, frequency tuning, noise, and stability. Antenna-based and large area emitters for Continuous-Wave (CW) or quasi-CW THz generation using two 1.55 μm lasers have been reported [6,7]. A Uni-Traveling Carrier Photodetector (UTC-PD), heterogeneously integrated on silicon, can provide a very high 3 dB bandwidth of more than 67 GHz with the responsivity of 0.7 A/W at 1.55 μm wavelength [8].

Recently, THz photonics wireless transmission link at 400 GHz was reported using On–Off Keying (OOK), Quadrature Phase-Shift Keying (QPSK), 16-Quadrature Amplitude Modulation (16-QAM), and 32-QAM based on photonics for THz generation [9]. A 64-QAM modulation format at 300 GHz

with a data rate of 0.096 Gb/s reported using UTC-PD. A 100 Gb/s transmission was demonstrated using a dual-polarization 16-QAM method in the W-band [10]. UTC-PD-based photonics at the transmitter and active Microwave Monolithic Integrated Circuits (MMICs) at the receiver was demonstrated experimentally at 237.5 GHz [11]. The 8-PSK modulation format was successfully transmitted at 240 GHz enabling a 30 Gb/s data rate capability [12]. Higher digital modulation formats are possible by integrating Mach–Zehnder Modulator (MZM) to one of the laser sources. A modified dual-drive In-phase and Quadrature (IQ) modulator driven by equal amplitude binary signals for generation of offset-free QPSK and 16-QAM constellations has been reported [13]. The MMW and THz coherent receivers rely on the availability of carrier phase-locking to the transmitter. Thus, after signal detection, digital synchronization (off-line demodulation) is required in order to construct the constellations (IQ map) and eye diagrams [11,12].

In this paper, we experimentally demonstrated an MMW wireless coherent communication link. The MMW link is based on an InGaAs p-i-n photomixer transmitter module and receiver. The modulated signal was generated using IQ-MZM with the required bitrate and output power. The proposed transmission link uses digital modulation formats and standard detection methods for W-band, which include horn antenna, W-band mixer, and Local Oscillator (LO) signal to downconvert the carrier wave from W-band to low Intermediate Frequency (IF) signal. The signal analysis of the IF signal was performed using a vector signal analyzer (VSA) software. Recently, we demonstrated On–Off Keying (OOK) modulation using photomixer and Schottky Barrier Diode (SBD) at the detection [14].

The remainder of the content for this paper arranged with sections and subsections, where Section 2 presents a general approach of the THz wave generation using photomixing. Sections 3, 3.1 and 3.2 present the experimental set-up with a detailed explanation of the block diagrams and the methods used to realize the MMW wireless communication link. In Section 4, the test results presented and discussed accordingly. Finally, in Section 5, a summary and future research plans described.

## 2. CW THz Wave Generation by Photomixing: General Approach

The generation of CW THz radiation by photomixing is based on the THz-periodic generation of electrons and holes in semiconductors by absorption of two interfering laser beams of frequencies $\nu_1$ and $\nu_2$. The interference results in the intensity modulation of the laser beams with the difference carrier frequency of $\nu_{THz}$. The two laser beams with electric field amplitudes of $E_1$ and $E_2$ focused onto an area of the semiconductor with dimensions smaller than the THz wavelength. The total electric field $E(t)$ yields

$$E(t) = E_1 \cdot \exp(i2\pi\nu_1 t) + E_2 \cdot \exp(i2\pi\nu_2 t) \tag{1}$$

Assuming the same polarization and amplitude of electrical field, $E_1 = E_2 = E_0$, the electrical field superposition on the photomixer can be rewritten as [7]

$$E(t) = E_0[\exp(i\pi(\nu_1 + \nu_2)t) \cdot \cos(\pi(\nu_2 - \nu_1)t)] \tag{2}$$

where $\nu_2 - \nu_1$ is the difference THz frequency, $\nu_{THz}$. The frequency component of $\nu_1 + \nu_2$ cannot pass the photomixer low-pass filter (LPF). The total laser power, $P_L$, where $P_L = P_1 + P_2$, becomes

$$\begin{aligned} P_L(t) = E^2(t) &= 2E_0{}^2[\cos(2\pi\nu_{THz}t) + 1] \\ &= P_0[1 + \cos(2\pi\nu_{THz}t)] \end{aligned} \tag{3}$$

where $P_1$ and $P_2$ are the laser sources output power. At photon energies above the band gap energy, the optical absorption coefficient of semiconductors is very large. As a result, the electron–hole pair generation takes place within a very thin layer compared to the THz wavelength. Electrons and holes are separated by a built-in field or an external field due to applied DC bias voltage. The resulting current contains a THz component due to the periodic modulation of the carrier generation rate.

The AC current is converted into THz radiation due to the photoconductive antenna, in which the device is connected. The ideal THz output power of antenna-based photomixer is given by [6,7]

$$P_{THz} = \frac{1}{2} R_A I_{THz}{}^2 = \frac{1}{2} R_A \left[ \frac{e P_L}{h\nu} \right]^2 \tag{4}$$

where $I_{THz}$ is the photogenerated current, $R_A$ is the radiation resistance of the antenna, $h\nu$ is the photon energy, and $e$ is electron charge. The radiation resistance is proportional to the wave impedance, $Z = \sqrt{\mu_0 \mu_{eff} / \epsilon_0 \epsilon_{eff}}$, where $Z_0 = \sqrt{\mu_0 / \epsilon_0} = 377 \, \Omega$ is the vacuum impedance and $\mu_{eff}$ & $\epsilon_{eff}$ are magnetic and dielectric constants, respectively. For non-dielectric materials, $\mu_{eff} = 1$, the radiation resistance is calculated as

$$R_A = \frac{60\pi}{\sqrt{\varepsilon_{eff}}} = \frac{60\pi}{\sqrt{\frac{1+\varepsilon_s}{2}}} \tag{5}$$

here, $\varepsilon_s$ is the dielectric constant. As shown in Equation (4), the quadratic relation between laser power and THz power is $P_{THz} = \gamma P_L{}^2$, where $\gamma$ is the scaling factor. The expression given in Equation (4) is a general form of the THz power emitted by an antenna with radiation resistance $R_A$ without including the limitations of the Resistor-Capacitor (RC) time constant and transport time of carriers. The THz power including the 3 dB transit time roll-off frequency, $v_{tr}$ and RC 3 dB roll-off frequency, $v_{RC}$ is given in Equation (6) [6]:

$$P_{THz} = \frac{1}{2} R_A I_{ph}{}^2 \left[ \frac{1}{1 + \left( \frac{v_{THz}}{v_{tr}} \right)^2} \right] \left[ \frac{1}{1 + \left( \frac{v_{THz}}{v_{RC}} \right)^2} \right] \tag{6}$$

The two frequencies $v_{tr}$ and $v_{RC}$ are calculated from $\frac{1}{2\tau_{tr}}$ and $\frac{1}{2\pi\tau_{RC}}$, respectively, where $\tau_{tr}$ is the transit time and $\tau_{RC}$ is the RC time constant of the device [6].

## 3. Experimental Setup

### 3.1. Optical and Receiver Channel

The optical channel of the MMW wireless communication link consists of two CW laser sources, IQ-MZM, Modulator Bias Control (MBC), Arbitrary Wave Generator (AWG), Erbium-Doped Fiber Amplifier (EDFA), Fused Fiber Polarization Combiner/Splitter, and optical fiber cables. The two independent laser sources used to generate the two optical signals are necessary for generating the required THz frequency difference $v_{THz}$. The first laser source is based on a Single-Frequency Laser (SFL), SFL1550P, from Thorlabs operating at $\lambda_1 = 1550.07$ nm. The SFL1550P source is current- and temperature-controlled, which allows tuning to specific wavelengths and power, which is essential to obtain the required MMW or THz-carrier frequency difference. The second laser source $\lambda_2 = 1549.35$ nm, is based on telecom modulation device, MX10B from Thorlabs. The MX10B is compatible with International Telecommunication Union (ITU) transmission channels in a 1.5 μm band. Measurements of the two lasers' exact wavelengths carried out using an Optical Spectrum Analyzer (OSA), AQ6331 from Yokogawa. The OSA is configured at 1549.73 nm central wavelength and 0.05 nm (5 GHz) wavelength resolution, which is the lowest resolution available for this device.

For the receiver channel, a high harmonic MMW mixer from Virginia's Diode, Inc. (VDI) integrated with MMW horn antenna of 23 dBi gain operating in the W-band used to downconvert the MMW signal into the low IF signal. The mixer has a 10 dB conversion loss at this frequency band. Using a stable signal synthesizer, the mixer LO is fed with 13.3333–14.6667 GHz frequency and 0 dBm to −3 dBm input power where the signal multiplied by 6 times to generate LO frequency in the range of 80 to 88 GHz. In this case, the received MMW signal and the LO signal produce the IF signal in the range of 2 to 10 GHz. The IF signal frequency, output power, noise, and the wireless link overall performance analyzed using Keysight Spectrum Analyzer (KSA) and processed by Vector Signal Analyzer (VSA) code, which performs real-time digital signal processing (DSP).

### 3.2. MMW Generator and Transmitter: InGaAs p-i-n Photomixer

As mentioned in the above section, the THz wave generation is implemented using different techniques and components. An antenna-integrated Low-Temperature Grown (LTG)-InGaAs photomixer, as used in this experiment Figure 1a, provides tunability, high output power, and wide bandwidth, i.e., above 2 THz, see Figure 1b. The photomixer is fiber-pigtailed and reverse-bias voltage-controlled with voltages between −2 V to 0.6 V, see Figure 1a. The DC voltage set point of −1.9 V corresponds to −11 mA DC photocurrent which gives maximum output power at 32.9 mW optical input power based on the device datasheet.

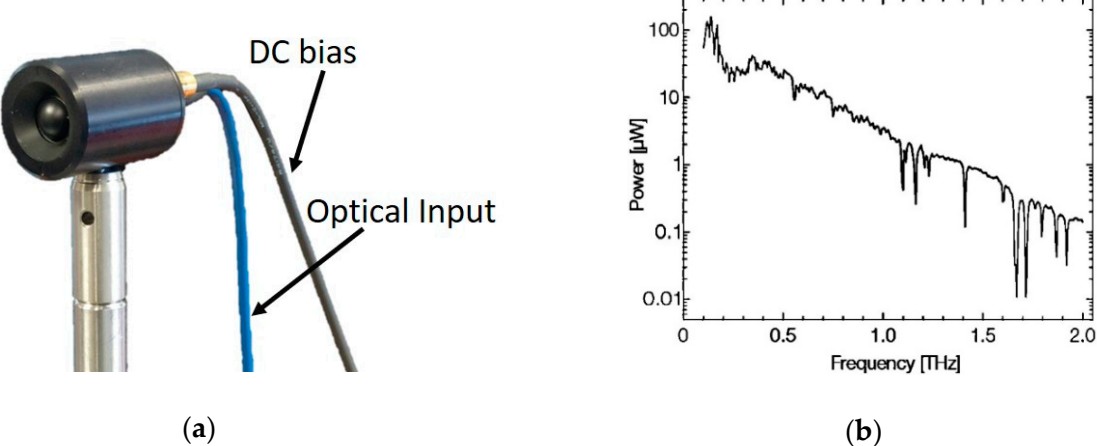

|        |        |
| :----: | :----: |
| (**a**) | (**b**) |

**Figure 1.** The millimeter-wave (MMW) generator and transmitter. (**a**) An InGaAs p-i-n photomixer module. (**b**) Output power spectrum of the photomixer [15].

The performance of the photomixer investigated using the two independent laser sources was configured with 90 GHz frequency difference. The combined optical power of the two lasers further amplified to 15 dBm and fed into the photomixer optical input. The photomixer antenna focused onto MMW horn antenna at a minimum distance. The received signal from the antenna was downconverted to low IF signal to analyze it using spectrum analyzer. Figure 2a shows the simplified schematic diagram used for evaluation of the photomixer.

The photomixer can radiate 65 μW output power at an input power of 32.9 mW, which means it has a conversion loss of ~27 dB around 100 GHz. We used the LO power, IF power, cable losses, and Friis transmission equation during the experiment to calculate the photomixer output power, which was found to be in good agreement with the manufacturer datasheet. As shown in Figure 2c, the photomixer generates a single CW frequency free of harmonic signals and spurious signals. After testing the performance of the photomixer as standalone, the proposed MMW wireless communication link built by including additional components. Figure 3 shows the proposed schematic diagram of the overall experimental set-up for QPSK MMW wireless communication link based on the photomixer.

As shown in Figure 3, the output of the MX10B laser fed into MXIQ-LN-30, an IQ-MZM modulator from iXblue Photonics, which operates at wavelengths of 1530–1580 nm with a maximum insertion loss of 7 dB at its maximum operating wavelength. It has a typical electro-optical bandwidth of 25 GHz and high input and output optical power capabilities. The optical output of the IQ-MZM connected to the modulator bias voltage and gain controller, where the dc bias voltage, gain, and dither signal adjustments were performed. The MBC enables configuring different operating points of the modulator by setting the bias voltages manually or automatically. A $2^7 − 1$ length Pseudorandom Binary Sequence (PRBS) signal with 50 Mbps data rate generated by Keysight Arbitrary Signal Generator (KAWG) was fed into the two IQ-MZM RF inputs. As a result, a QPSK modulated signal of 100 Mbps bitrate was achieved at the output of the MBC. As a result, a QPSK modulated signal of 100 Mbps bitrate was achieved at the output of the MBC. The modulated optical signal output at the MBC and the SFL were

combined using PFC1550F, and amplified using EDFA100P to 15 dBm, which is required at the input of the photomixer to get maximum output power.

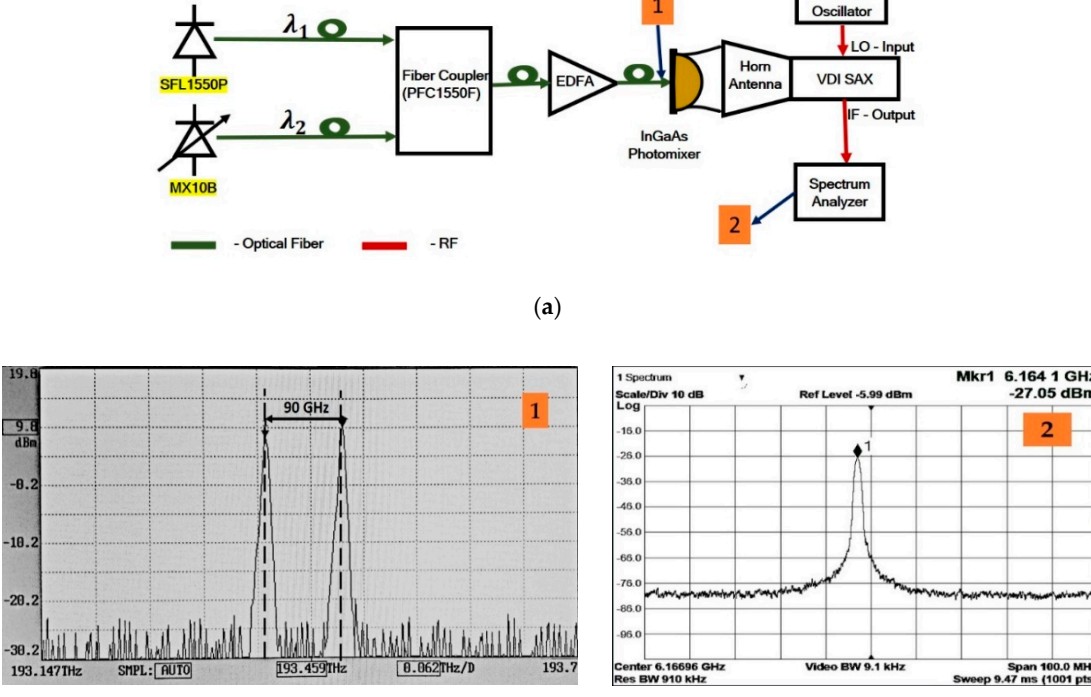

**Figure 2.** Photomixer evaluation test set-up. (**a**) A schematic diagram of the test set-up with the optical link and receiver channels. (**b**) Two optical signals at the photomixer input with 90 GHz frequency difference. (**c**) Downconverted infrared (IF) signal measured by Keysight Spectrum Analyzer (KSA).

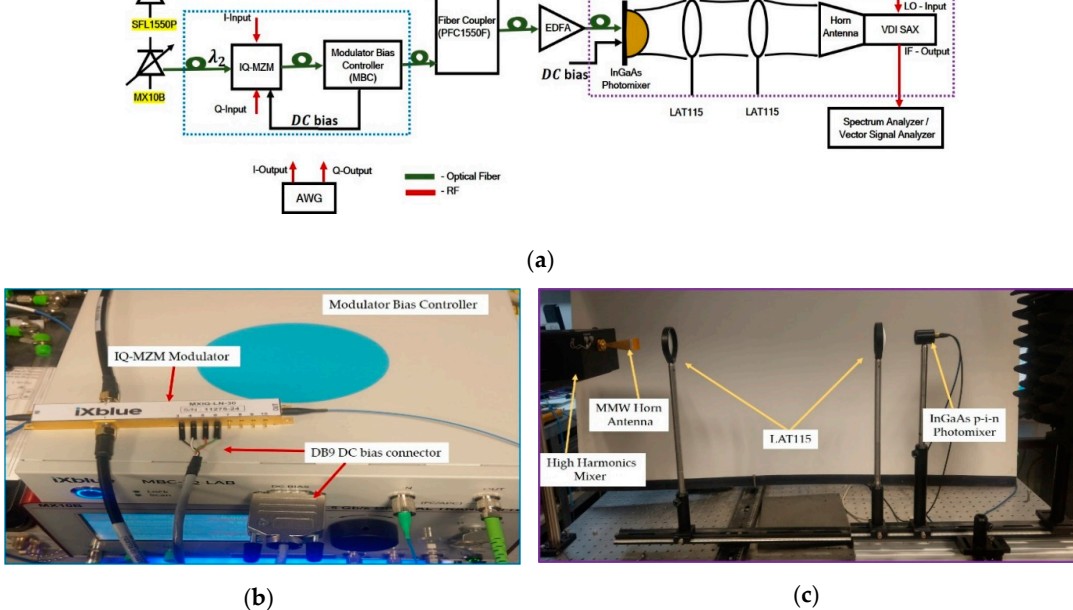

**Figure 3.** Experimental set-up. (**a**) Proposed MMW wireless communication link based on InGaAs p-i-n photomixer and receiver. (**b**) IQ-MZM modulator and the Modulator Bias Control (MBC). (**c**) The free space transmission overview.

The radiated signal from the photomixer focused to the direction of the receiver antenna by two identical LAT115 lenses. The two LAT115 are plano-convex Polytetrafluoroethylene (PTFE) lenses with 76.2 mm and 115 mm of diameter and focal length, respectively, at 500 GHz. The carrier wave can be focused on a longer distance by placing the lenses in a different configuration. The 90 GHz CW received at a fixed distance of 0.5 m between the photomixer and the horn antenna. The received signal was further downconverted to IF signal using the high harmonic mixer as done in the valuation of the photomixer. The downconverted signal was further analyzed using KSA and VSA, where the demodulation results, i.e., the constellation, eye pattern, and Bit Error Rate (BER) calculation, were achieved.

## 4. Experimental Results and Discussion

The generated MMW was first analyzed using KSA to measure the magnitude and frequency of the new test set-up, see Figure 3. The VSA demodulator was configured with the center frequency and symbol rate using the parameters acquired from the KSA and the input data rate. As described in the above section, after analyzing the CW signal, the PRBS signals were added to the RF inputs of the IQ-MZM. The modulator can tolerate RF input signals of 6 V peak to peak. In this experiment, PRBS signals with peak to peak voltages of 1 V, 1.5 V, and 3.5 V were applied to the modulator RF inputs. A real-time received QPSK signal reconstruction for the different peak to peak voltage values of the PRBS signals is presented in the following pictures.

The IQ constellations of the received signal for the 1 Vpp PRBS input signal are more scattered and non-symmetrical. As can be seen in Figure 4b, the opening of the eye diagram does not appear to be broad and clear. The symbols are not fully detected due to low bit energy at the transmitter.

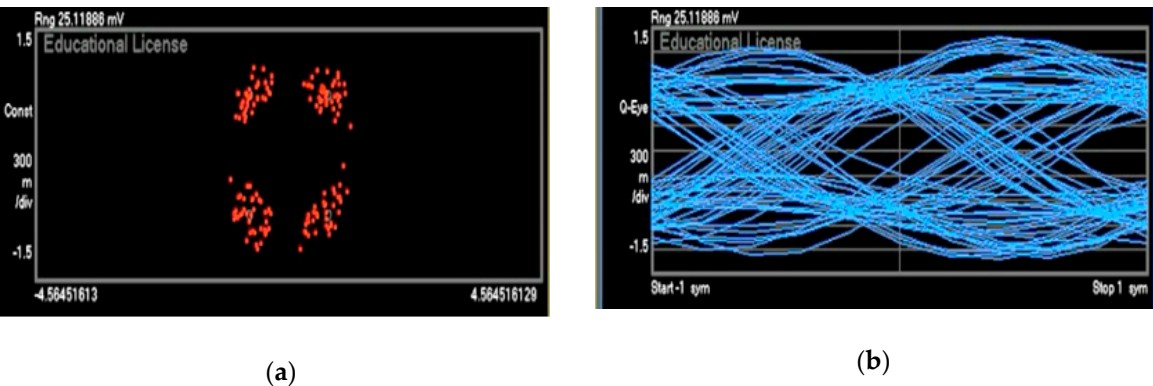

(**a**)　　　　　　　　　　　　　　　　　　　　　　　　　(**b**)

**Figure 4.** Real-time received QPSK signal reconstruction using VSA code for transmitted 1 Vpp PRBS signal: (**a**) IQ constellation and (**b**) eye pattern.

When the signal power was increased by 0.5 Vpp, as shown in Figure 5, the IQ constellations and the eye diagram improved compared to the 1 V pp PRBS signals. The symbols are close to the ideal IQ symbol placements and the symmetry of the symbols improved. The eye diagram opening is broad and clear without a symbol crossing. By further increasing the bit energy of the transmitted signal, the constellation and the eye diagram measurements can show an improvement. Figure 6 shows an open and clean eye diagram with a better symmetry of the symbols for 3.5 Vpp PRBS signal input.

As shown in the above figures, the IQ constellation and the eye pattern improved as the data voltage increased; thus, the QPSK signal was detected more precisely. The estimations of the BER and Signal-to-Noise Ratio (SNR) can be done using the Q-factor equations found in the works in [16] or [17], based on the given eye diagrams. Table 1 presents the Q-factor calculated values, BER, $\frac{E_b}{N_0}$, bit rate, bit period, and timing jitter depending on the peak to peak voltage values of the input data.

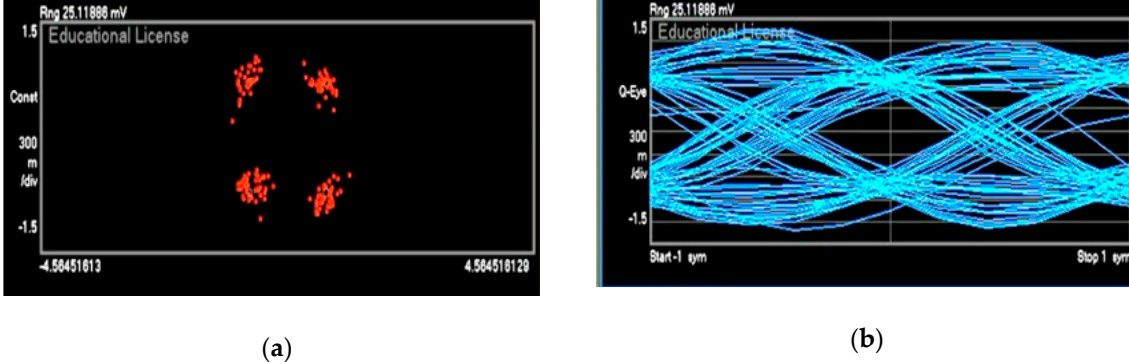

(**a**)                                                            (**b**)

**Figure 5.** Real-time received QPSK signal reconstruction using VSA code for transmitted 1.5 Vpp PRBS signal. (**a**) IQ constellation and (**b**) eye pattern.

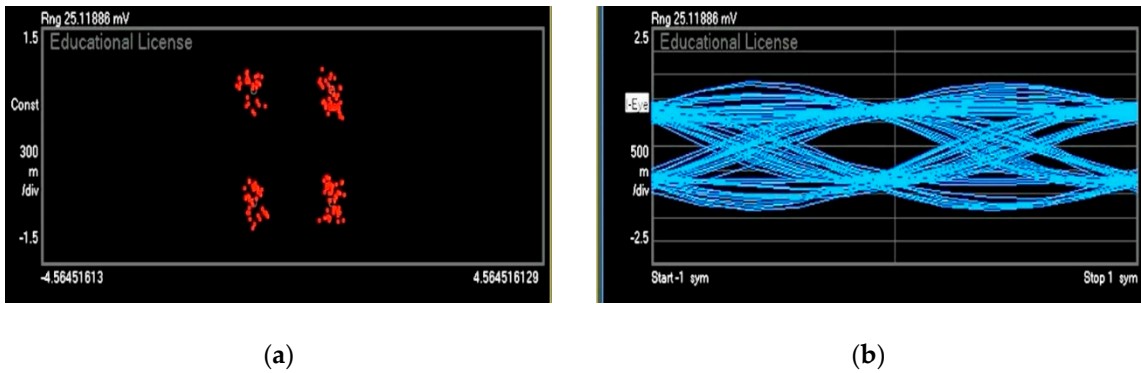

(**a**)                                                            (**b**)

**Figure 6.** Real-time received QPSK signal reconstruction using VSA code for transmitted 3.5 Vpp PRBS signal. (**a**) IQ constellation and (**b**) eye pattern.

**Table 1.** Calculated values of the Q-factor and bit error rate (BER) for deferent peak to peak voltage values.

| Input Data Voltage (Vpp) | Q-Factor | BER | $E_b/N_o$ (dB) | Bit Rate (Mbps) | Bit Period (ns) | Timing Jitter (ns) |
|---|---|---|---|---|---|---|
| 1 | 1.115 | $1.32 \times 10^{-1}$ | −2.05 | 100 | 10 | 6.27 (62.72%) |
| 1.5 | 1.411 | $7.91 \times 10^{-2}$ | 0 | 100 | 10 | 5.25 (52.57%) |
| 3.5 | 3.667 | $1.23 \times 10^{-4}$ | 8.286 | 100 | 10 | 4.15 (41.5%) |

The calculated BER value for 3.5 Vpp PRBS input data, as given in Table 1, is in the order of one magnitude better than the Hard-Decision Forward Error Correction Threshold (HD-FEC) of $3.8 \times 10^{-3}$. For the same test, an Error Vector Magnitude (EVM) value of 20% rms was achieved. By applying different modern error-correcting algorithms, the BER value can further improved.

The state-of-the-art technologies for MMW communication are summarized in Table 2. The table compares researches based on the carrier wave frequency, data rate, achieved BER or EVM value, demodulation method, and the maximum transmission distance. Even though other reported researches are available, only the researches with a carrier frequency of less than 200 GHz presented.

**Table 2.** Summarization and comparison of reported MMW wireless communication link researches with carrier wave frequency under 200 GHz.

| Reference | Carrier Frequency (GHz) | Data Rate (Gb/s) | BER/EVM | Demodulation Method | Transmission Distance (m) |
|---|---|---|---|---|---|
| [4] | 100 | 100 | $<3.8 \times 10^{-3}$ | Off-line DSP | 0.7 |
| [10] | 87.5 | 100 | $10^{-3}$ | Off-line DSP | 1.2 |
| [17] | 196 | 0.1 | EVM < 10% | Same Tx/Rx | 0.5 |
| [18] | 120 | 10 | $<10^{-10}$ | Real time | 200 |
| [19] | 146 | 1 | -* | Off-line DSP | 0.025 |
| [20] | 200 | 1 | $<10^{-9}$ | Real time | 2.6 |
| This work | 90 | 0.1 | $1.23 \times 10^{-4}$ | Real time | 0.5 |

* The value is unavailable.

The generation of the carrier wave and detection methods for the realization of MMW wireless communications of the reported works in Table 2 is done using different approaches. In the W-band, at 87.5 GHz and 100 GHz, experimental demonstration of wireless communication links was reported using photonic upconversion of a polarization multiplexed 16-QAM optical baseband signal and an optical wireless integration system delivering two-channel Polarization Division Multiplexing Quadrature Phase Shift Keying (PDM-QPSK), respectively [4,10]. Both of the experiments achieved close BER performance for 100 Gb/s data rate transmission. In the THz coherent receiver, where a phase-locking between the transmitter and the carrier wave is required, the homodyne configuration can be used. This configuration uses the same reference signal for the generation or detection of the THz modulated waves. Using this approach, a transceiver operating at 196 GHz carrier frequency based on split-block waveguide integrated microwave assemblies, with MMICs realized using advanced InP HEMTs with sub-50 nm gate lengths [17]. In this case, an EVM of below 10% rms achieved for a transmission distance of 0.5 m. The reported works in [18,19] used a photonic 1.55 μm approach based on UTC-PD, driven by standard laser lines, and dedicated dual-mode lasers. The 1.55 μm photomixer composed of a UTC-PD monolithically integrated with a wide bandwidth antenna was used for the generation of 200 GHz carrier frequency with a receiver based on a heterodyne mixing and amplitude modulation detector achieved 1 Gb/s data rate for 2.6 m free-space transmissions [20]. Due to the lack of fast real-time demodulators, as the data rate increased, most reported works used off-line demodulation processes. Compared to the reported methods in Table 2, using an InGaAs photomixer module for the generation of the carrier wave and heterodyne mixing at the receiver with a real-time DSP gives a better BER value for 100 Mb/s data rate transmission at a distance of 0.5 m. In contrast to other works, where dedicated laser sources, demodulation methods, and carrier wave generators are required, this work achieved better performance based on standard telecom and customer available components.

## 5. Conclusions

We have demonstrated QPSK MMW wireless communication based on an off-the-shelf InGaAs p-i-n photomixer. The experiment showed the capabilities of the photomixer for wireless communication use. Throughout the experiment, we used only customer available components, which represents a promising solution for 5th generation wireless communications, IoT, fast point-to-point communication for autonomous driving, and usage on available optical telecom infrastructures. Here, we showed the transmission of a 100 Mbps data rate QPSK modulated signal, even though the photomixer has a 2 THz bandwidth, which can used for higher data rates and advanced modulation formats. As shown in the above results, wireless communication based on customer available photomixer and optical telecom infrastructures is more feasible in the near future. In the current experiments, we have shown results of low data rate and modulation scheme based on the available lab equipment.

In future work, further research will be done based on the photomixer for higher data rates and advanced formats, such as M-QAM and Orthogonal Frequency-Division Multiplexing (OFDM) formats.

The work will include a detailed characterization of the wireless channel link for different formats, a long-distance wireless transmission as shown in this paper, and applying different FEC techniques to improve the performance of the wireless link. Assuming an Additive White Gaussian Noise (AWGN) channel and FEC based on Reed–Solomon code, i.e., RS(255,239) as used in undersea lightwave systems [21] and Optical Transport Network (OTN) standards [22], which is the first-generation FEC [23], input BER of $10^{-4}$ can be improved to post-FEC BER of $10^{-12}$ with 7% overhead. This algorithm can provide approximately 6 dB Net Coding Gain (NCG), very significant performance improvement in high-speed wireless networks. In addition to this type of FEC, more powerful FEC codes such as turbo codes, Turbo Product Codes (TPC), and Low-Density Parity-Check Codes (LDPC) are already developed [24]. For example, the LPDC in [25] can correct a BER of $10^{-3}$ to $10^{-12}$ with an overhead of 7%, and the Bose–Chaudhuri–Hocquenghem (BCH)-based turbo code can correct a BER of $10^{-2}$ to $10^{-12}$ with an overhead of 21.5% [23]. The different FEC algorithm encoders and decoders will be realized using Field-Programmable Gate Array (FPGA) implementations.

**Author Contributions:** Conceptualization, A.A.; formal analysis, A.W.; investigation, A.W.; writing—original draft preparation, A.W.; validation, A.W., A.A., and D.R.; data curation, A.W.; resources, A.A., D.R., and A.W.; writing—review and editing, A.W., A.A., and D.R.; supervision, A.A.; funding acquisition, A.A.; All authors have read and agreed to the published version of the manuscript.

**Funding:** This work was partially supported by the Ministry of Science and Technology, Israel.

**Conflicts of Interest:** The authors declare no conflicts of interest.

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
