# Peer review of "QPSK MMW Wireless Communication System Based On p-i-n InGaAs Photomixer"

_electronics, doi:10.3390/electronics9081182_

Round 1
Reviewer 1 Report
The manuscript is centering a hot topic and as such it provides a nice solution. A few hits to improve the readability and quality of the presentation.
Sub-sections are needed all along the manuscript. Many infos are provided flooding freely, while they merit a much more detailed description. In particular the photomixer details and stand-alone performance could be presented separately, since it is the key element of the setup.
Figures regarding the measured performance are too small and should be enlarged and presented/discussed separately (figs 4, 5 and 6).
finally, alternative approaches are cited but not compared quantitatively. This should be the case after presenting the measured results, even in tabular format.
Some work on spelling and grammar necessary.
Author Response
Please see the attachement.

Reviewer 2 Report
The authors propose an experimental demonstrator for wireless coherent communications in the millimeter wave (MMW) band. The proposed system is based on two laser sources, an IQ-MZM modulator, a fiber coupler, and a photo-mixer for the transmitter side, and horn antennas, W-band mixer and local oscillator for receiver side. The carrier wave is down-converted to low IF signal, which is further analyzed using vector signal analyzer VSA) software.
The paper is not a fundamental research one, but rather an implementation one.
The paper is well written, the scope and objectives are matched with the journal, the experimentation methodology is adequate, and the results are satisfactory.
However, I have some comments/suggestions for improving the work, as described below.
- Paper title contains “… based on … and …”, which means that the proposed communication system is based on an InGaAs photomixer AND a telecommunication infrastructure? I think you must replace “and” with “for”, or, for simplicity, you can delete “and telecom infrastructure”
- Section 2: Please reformulate some sentences for a clear presentation. You can use distinct paragraphs to describe the transmitter, the optical channel and the receiver.
- “Optical output of the IQ-MZM connected to modulator bias voltage and gain controller (MBC) which the dc bias voltage, gain and dither signal are controlled.”, “… which gave 100 Mbps bitrate QPSK modulated signal at the output of the MBC. The modulated optical signal output at the MBC and the SFL are combined …”
The above description does not correspond to fig.1, where the optical output (green arrow) from IQ-MZM is fed directly to the fiber coupler not to the MBC. The MBC controls the operation of IQ-MZM using DC signals (black arrow) through the DB-9 connector.
But, according with the MBC-IQ-LAB-A1 datasheet (https://photonics.ixblue.com/sites/default/files/2018-12/IQ%20Modulator%20Bias%20Controller_2.pdf), the output of IQ modulator can be connected to the input of the MBC, and the output of MBC to the fiber coupler. If this is your case, please modify fig.1 by inserting the MBC between IQ-MZM and fiber coupler.
- “The two IQ-MZM RF inputs were modulated using …”
The optical output from IQ-MZM is modulated, not the inputs. Maybe, “The two RF inputs are generated using …”
- A picture with the proposed testbed and all measurement instruments can be also included.
- fig.2 can be replaced using a print screen option from the menu of the instrument, not using a phone camera? Figs. 3-6 are ok.
- the performance of a communication system is usually described in terms of BER(SNR) graphs.
For all considered peak-to-peak voltages, you can compute the SNR from the Q-factor, and further the BER? Draw a table or graph. Other measurements can be derived for all eye diagrams: SNR, jitter, bit period, bit rate, etc?
- To simulate a communication channel with variable SNR, two scenarios are available: 1. variable transmit power and fixed noise power (fixed distance between TX and RX), which is your case; 2. fixed transmit power and variable distance between TX and RX. Please analyze also the second case.
- Enter in the Conclusions section future development directions starting from presentation of FEC techniques in previous section. How can you extend the proposed system to implement FEC coding and decoding? Maybe using FPGAs?
- Correct the typos, like:
-- using dots after equations.
-- using commas:
--- pag.2: “… of electrical field, ?1=?2=?0, the electrical field superposition …”
--- after equations or before “where”/”which”: “… equation 3, where P1 and P2 …”, “… equation 4, where ITHz is …”, “The two frequencies miutr and miuRC are …, where tautr is …”, “…it was set at -1.9 V, which corresponds …”, “… the -3 dB input power, where the signal is multiplied by 6 times in order to … ”, “For the same test, EVM value …”
-- acronyms must be marked with Upper-Case in text: “A uni-traveling carrier photodetector (UTC-PD)” -> “An Uni-Traveling Carrier Photodetector (UTC-PD)”, “low pass filter (LPF)” -> “Low Pass Filter (LPF)”. Other acronyms are QPSK, QAM, MMIC, VSA, SFL, KAWG, OSA, IF, BER, HD-FEC, NCG, OTN, TPC,
-- other acronyms must be defined: AWGN, EVM
-- use the verb “is”: “… where Z0 =…= 377 ohms is the …”, “The MX10B laser out fed into MXIQ-LN-30 an IQ-MZM from …” -> “The output of MX10B laser is fed into MXIQ-LN-30, an IQ-MZM modulator from …”, “The OSA is configured at … resolution, which is …”, “… the mixer LO is fed with 13.3333 GHz …”etc.
-- one space between number and measuring unit: “1.55micro”, “50Mbps”, “1V, 1.5V and 3.5V ”
-- titles of figs 4-6: peak-to-peak voltage is abbreviated Vpp not Vptp
-- “As shown on Eq. (4)” -> “As shown in Eq. (4)”
-- “The generated MMW was firstly analyzed …”
Author Response
Please see the attachement.

Reviewer 3 Report
Wireless transmission of digital data in THz and mmW-bands was instrumented and observed by the authors as others do.
As to novelty and substatntial progress, nothing new is found from the methodology to the configuration of the system.
- Equations are written, but they are common.
- Figure 1 is not new.
- The authors should have brought pictures of the componenets and system of Figure 1.
- Figure 2 is not proper for journal paper writing. It should be put in a right format via a professional plotting program. The photograph is an extra thing.
- Figure 3 is not proper for journal paper writing. It should be put in a right format via a professional plotting program. The photograph is an extra thing.
- Figures 4, 5, and 6 have eye-diagrams as pictures. Really hard to see quantities of the eye-opening there.
Please, be aware that there are many mistakes of the English usage like the followings
In the abstract,
'is promising solution '
'at MMW band'
In p.2
'due to large optical absorption coefficient of semiconductors '
In p.3
'MMW and THz coherent receivers relay on the availability'
In p.3
'The MX10B laser out fed into MXIQ-LN-30 an IQ-MZM from iXblue Photonics which operates at wavelengths of 1530 – 1580 nm with maximum insertion loss of 7 dB at its maximum operating wavelength'
In p.3
'Block diagram of the proposed coherent communication link is shown'
Author Response
Please see the attachement.

Round 2
Reviewer 1 Report
The alternative communication link proposals should be commented, after being listed and reported in Table 2
Author Response
Dear Reviewer 1,
The authors would like to thank you for the additional remarks. An additional paragraph is dedicated to reported researches proceeding table 2.
comment:
- The alternative communication link proposals should be commented, after being listed and reported in Table 2
Response:
- An additional paragraph included on page 8, lines 223 – 245.